# Assessment of Phenotypic Variations and Correlation among Seed Composition Traits in Mutagenized Soybean Populations

**DOI:** 10.3390/genes10120975

**Published:** 2019-11-27

**Authors:** Zhou Zhou, Naoufal Lakhssassi, Mallory A. Cullen, Abdelhalim El Baz, Tri D. Vuong, Henry T. Nguyen, Khalid Meksem

**Affiliations:** 1Department of Plant, Soil, and Agricultural Systems, Southern Illinois University, Carbondale, IL 62901, USA; zhou1228@siu.edu (Z.Z.); naoufal.lakhssassi@siu.edu (N.L.); mallory.cullen@siu.edu (M.A.C.); abdelhalim.elbaze@siu.edu (A.E.B.); 2Division of Plant Sciences, University of Missouri, Columbia, MO 65211, USA; vuongt@missouri.edu (T.D.V.); nguyenhenry@missouri.edu (H.T.N.)

**Keywords:** TILLING, mutation breeding, chemical mutagenesis, soybean meal traits, soybean oil composition traits, correlation analysis

## Abstract

Soybean [*Glycine max* (L.) Merr.] seed is a valuable source of protein and oil worldwide. Traditionally, the natural variations were heavily used in conventional soybean breeding programs to select desired traits. However, traditional plant breeding is encumbered with low frequencies of spontaneous mutations. In mutation breeding, genetic variations from induced mutations provide abundant sources of alterations in important soybean traits; this facilitated the development of soybean germplasm with modified seed composition traits to meet the different needs of end users. In this study, a total of 2366 ‘Forrest’-derived M2 families were developed for both forward and reverse genetic studies. A subset of 881 M3 families was forward genetically screened to measure the contents of protein, oil, carbohydrates, and fatty acids. A total of 14 mutants were identified to have stable seed composition phenotypes observed in both M3 and M4 generations. Correlation analyses have been conducted among ten seed composition traits and compared to a collection of 103 soybean germplasms. Mainly, ethyl methanesulfonate (EMS) mutagenesis had a strong impact on the seed-composition correlation that was observed among the 103 soybean germplasms, which offers multiple benefits for the soybean farmers and industry to breed for desired multiple seed phenotypes.

## 1. Introduction

Soybean [*Glycine max* (L.) Merr.] is one of the most valuable crops worldwide, mainly for its high protein meal and oil content. In 2017, soybean accounted for 70% of global protein meal consumption and 53% of vegetable oil consumption in the United States of America (USA) [1]. The soybean seed composition is typically comprised of 40% protein, 21% oil, and 11% soluble carbohydrates, on a dry weight basis [2]. In soybean meal, soy protein is not only the main source of livestock feeds, but has also been increasingly utilized for soyfood products, such as tofu and soymilk [3]. Soluble carbohydrates in the seed are another important component of soybean meal and provide metabolizable energy in the diets of monogastric animals. The disaccharide sucrose (2.5%–8.2%) and the two oligosaccharides raffinose (0.1%–0.9%) and stachyose (1.4%–4.1%) are the three major soluble carbohydrates present in soybean seed [4]. Sucrose is regarded as a beneficial ingredient in livestock feeds, whereas the raffinose family oligosaccharides (RFOs), for example, raffinose and stachyose, are undesirable polysaccharides. RFOs have anti-nutritional effects on soybean meal, which causes poor digestion in monogastric animals [5]. Soybean oil has been widely used in salad dressings, cooking oils, plastics, and biodiesel fuel. It is comprised of five principal fatty acids, including palmitic acid (16:0, 10%), stearic acid (18:0, 4%), oleic acid (18:1, 22%), linoleic acid (18:2, 54%), and linolenic acid (18:3, 10%) [6]. Reducing palmitic acid content in soybean oil can reduce the risk of coronary heart disease associated with saturated fats, while stearic acid improves oxidative stability of soybean oil and has no negative effect on blood serum low-density lipoprotein (LDL) cholesterol [7,8]. Elevated oleic acid concentrations in soybean oil are desirable for human consumption due to the stability at high cooking temperatures and extended shelf life of oleic acid [9]. Although the soybean oil with high linoleic acid content has low oxidative stability, alpha-linolenic acid is the precursor of eicosapentaenoic acid (EPA, 20:5) and docosahexaenoic acid (DHA, 22:6), which are important omega-3 fatty acid supplements for human health [10]. Researchers achieved modification of the fatty acid content in soybean seed through targeting catalytic enzymes in the fatty acid biosynthetic pathways, such as 3-ketoacyl-acyl carrier protein (ACP) synthase II (KASII), 16:0-ACP fatty acid thioesterase (FATB), the omega-6 fatty acid desaturase 2 (FAD2), and the omega-3 fatty acid desaturase (FAD3) [11,12,13,14].

In plants, the carbon distribution amongst the different metabolic pathways in the carbon cycle determines the levels of various chemical compounds produced, such as amino acids, fatty acids, and nucleic acids [15]. Although elevated protein, oil, and sucrose content increases the value of soybeans for a variety of applications, it is extremely difficult to develop a soybean cultivar with a high content of all of those traits due to the limited amount of carbon available for all of the pathways. A strong negative correlation between protein and oil content has been previously reported by numerous studies [16,17,18]. Protein content was also found to negatively correlate with seed sucrose content, whereas protein and stachyose contents were positively correlated [4]. Among soluble carbohydrates, the positive correlation between sucrose and raffinose content added an obstacle for breeders to increase sucrose content in soybean cultivars while remaining low in raffinose content [19]. In addition, correlations among the five principal fatty acids in soybean revealed that oleic acid content was negatively correlated with stearic acid, linoleic acid, and linolenic acid content due to the fluctuation within the fatty acid biosynthetic pathway [20,21]. To some extent, these relationships among seed composition traits confine the potential of developing soybean germplasm with multiple desired traits. 

Given the existing relationships among seed composition traits from elite, domesticated, and semi-domesticated soybean germplasm lines, breeders could hardly develop cultivars with multiple desired traits. Mutation breeding is increasing in popularity among crop breeding programs due to the wide range of genetic diversity that can be induced using physical and chemical mutagens, such as X-rays and ethyl methanesulfonate (EMS). Furthermore, mutation breeding offers the benefit of not introducing foreign DNA as a form of genetic diversity, preventing the restriction and utilization of mutant plants when compared to genetically engineered crops [22]. Currently, more than 3200 mutant varieties have been released from 214 plant species throughout the world [23]. The efficient reverse genetics method to determine the function of gene interests in plants, TILLING (Targeting Induced Local Lesions IN Genomes), has been applied to identify the induced mutations from diverse mutagenized populations [24,25,26,27,28]. TILLING typically entails chemical mutagenesis and a high-throughput mutation screening method [29]. The first soybean TILLING platform was established in 2008, in which four independent mutagenized populations were produced with a range of mutation densities from ~1/140 kb to ~1/550 kb using both EMS and N-nitroso-N-methylurea (NMU) [30]. Since then, multiple soybean mutant libraries were developed for trait improvement and functional gene analyses [31,32,33,34]. However, in most cases, raw data from seed composition traits including protein, oil, fatty acids, and carbohydrates were not available to perform the correlation analysis. The objective of this study was to develop an EMS mutagenized soybean population for seed composition phenotypic assessment and correlation analysis among different pairs of traits. 

## 2. Materials and Methods 

### 2.1. EMS Mutagenesis and Mutant Population Development

A full-scale mutagenesis test was carried out at ten different concentrations of EMS ranging from 0% to 1.0% (v/v) (0%, 0.3%, 0.4%, 0.5%, 0.55%, 0.6%, 0.65%, 0.7%, 0.8%, 1.0% EMS solutions) to determine the EMS concentration that should be used for the development of a large mutant population. Each set of one hundred soybean (*Glycine max*) seeds from the ’Forrest’ cultivar was soaked with different concentrations of EMS in 500 mL bottles at room temperature overnight (15 h) in a fume hood. Afterwards, the seeds were washed thoroughly three times using 300 mL of water per wash to remove excess EMS from the seeds. Rinse water was neutralized with a 10% (w/v) sodium thiosulfate solution. The mutagenized seeds were sowed immediately in 48-cell trays filled with ProMix BX (Premier Tech., Rivière-du-Loup, Québec, QC, Canada). Seeds were grown in a greenhouse at the Horticulture Research Center (HRC) at Southern Illinois University Carbondale under a 16 h light/8 h dark photoperiod at 28–30 °C. After 10 days, the germination rates of each treatment were recorded. The concentration of EMS used to mutagenize the large ‘Forrest’ population was calculated from the treatment showing a lethal dose (LD_50_) when compared to the wild-type [35].

In the first season, a total of 8000 ‘Forrest’ seeds were soaked in 0.64% (v/v) EMS solution overnight (15 h) at room temperature in the fume hood followed by three thorough washes with water the next day. The mutagenized seeds (M1) were then planted in the 48-cell trays. Seedlings were grown in a greenhouse under a 16 h/8 h light/dark photoperiod at 28–30 °C. After 3–4 weeks, the seedlings were transplanted to the field and the M1 plants produced M2 seeds through self-pollination. M2 seeds were harvested, packaged, and stored at −20 °C in the fall. During the second spring season, M2 seeds were sown by single-seed descent in the greenhouse and the seedlings were transplanted into the field after 4 weeks. Young leaf tissue from each M2 plant was collected in the field for DNA extraction, and the M3 seeds were harvested, threshed, and stored for phenotyping.

### 2.2. Seed Protein, Oil, and Carbohydrates Phenotyping

Total protein and oil content in M3 and M4 mutant lines were quantified using FOSS near-infrared reflectance (NIR) spectroscopy system Model 6500 (FOSS, Nils Foss Alle 1, DK-3400 Hilleroed Denmark). Approximately 5–6 g of whole soybean seed were processed for quantification following a previously reported procedure [36] with a minor modification as described by Pathan et al. [37]. Three major components of soybean carbohydrates of the M3 and M4 mutants were quantified using the high performance liquid chromatography (HPLC) instrument (Agilent, Santa Clara, CA, USA) following a simple analytical method as previously described by Valliyodan et al. [38]. Briefly, approximately 1.0 g of soybean seeds was ground with a 20-mesh screen. The powder was lyophilized for two days followed by an extraction procedure. The resulting extract was used for HPLC assays along with pre-prepared sugar standards. 

### 2.3. Seed Fatty Acids Phenotyping

For M3 mutant lines, a five-seed sample taken from each mutant line was placed in an envelope and manually crushed with a hammer. Crushed seeds were extracted in 5 mL chloroform:hexane:methanol (8:5:2, v/v/v) overnight. Derivitization was done by transferring 100 μL of extract to a vial and adding 75 μL of methylating reagent (0.25 M methanolic sodium methoxide:petroleum ether:ethyl ether, 1:5:2 v/v/v). Hexane was added to dilute samples to approximately 1 mL. An Agilent (Palo Alto, CA, USA) series 6890 capillary gas chromatograph fitted with a flame ionization detector (275 °C) was used with an AT-Silar capillary column (Alltech Associates, Deerfield, IL, USA). Standard fatty acid mixtures (Animal and Vegetable Oil Reference Mixture 6, AOACS) were used as calibration reference standards. Percent palmitic, stearic, oleic, linoleic, and linolenic acid contents in the oil were determined. Five major fatty acid contents were also measured from selected M4 lines according to the two-step methylation procedure [39]. At least three seeds per line were individually crushed in 16 mm × 200 mm tubes with Teflon-lined screw caps. Then, 2 mL of sodium methoxide was added into each tube followed by 50 °C incubation for 10 min. After 5 min of cooling, the samples were mixed with 3 mL of 5% (v/v) methanolic HCl, incubated at 80 °C for 10 min, and cooled for 7 min. Then, 7.5 mL of 6% (w/v) potassium carbonate and 2 mL of hexane were added to each tube and centrifuged at 1200 g for 5 min. The upper layers were transferred to vials, from which the individual fatty acid contents were determined as a percentage of the total fatty acid content in soybean seed by gas chromatography. A Shimadzu GC-2010 (Shimadzu Co., Kyoto, Japan) gas chromatograph, fitted with a flame ionization detector, was equipped with a 60 m SP-2560 fused silica capillary famewax column (0.25 mm i.d. × 0.25 μm film thickness) (Supelco, Inc., Bellefonte, PA, USA). Standard fatty acids (Nu-Chek-Prep., Elysian, MN, USA) were run first to create a calibration reference.

### 2.4. Statistical Analysis

Distribution, One-way ANOVA and Student’s t test were performed for the phenotype results from selected individual mutant lines using JMP14 software (SAS Institute Inc., Cary, NC, USA). Various descriptive statistics were calculated using PROC UNIVARIATE, in which the normal quantile plots were generated by adding the command qqplot with the normal option to the program. Phenotypic correlation analysis was conducted using the correlation procedure (PROC CORR) in the SAS9.4 software (SAS Institute Inc., Cary, NC, USA).

## 3. Results

### 3.1. Development of Chemically Mutagenized Soybean Populations

The calculated 0.64% (v/v) EMS concentration that generated an approximate 50% germination rate was used to treat a total of 8000 ‘Forrest’ wild-type seeds (Figure 1A; Appendix A). In the first planting season, all germinated M1 seedlings were transplanted from the greenhouse into the field, resulting in 2366 M1 plants that were harvested to collect the M2 seeds. In the second planting season, M3 seeds were harvested and stored at −20 °C to constitute a mutant seed bank for seed phenotypic analysis (Figure 1A). EMS mutagenesis resulted in many mutants presenting a wide range of morphological phenotypes, including multiple leaflets at a single node, whereas most wild-type soybean leaves are trifoliolate except the first two unifoliolate leaves (Appendix A). Some mutants had 3–4 branches with a large number of pods, while a few other plants were found to present a compact phenotype or to be dwarf mutants. Other observed phenotypes included leaf lesions, leaf color, leaf shape, leaf texture, vine-like, and chlorotic leaves (Appendix A).

### 3.2. Seed Composition Phenotypes of Mutagenized Soybean Populations

A subset of 881 M3 lines were assayed to measure the seed composition traits, including protein, oil, carbohydrate, and fatty acid content (Figure 2 and Figure 3). For the carbohydrate phenotype, mutants exhibited a wide range of sucrose content (0.4%–8.6%) and stachyose content (0.5%–6.2%) (Appendix A). One M3 mutant showed relatively high sucrose content (8.1%) and low RFOs content (0.5% raffinose and 0.5% stachyose) when compared to the wild-type cultivar, ‘Forrest’. The maximum values of protein and oil content from the M3 mutants reached 41.6% and 20.7%, respectively (Appendix A). In comparison with the wild-type, one M3 mutant presented both high protein content (39.3%) and oil content (20.7%). Moreover, the ranges of the five principal fatty acids showed significant variations among the M3 mutants, including palmitic acid (7.2%–13.9%), stearic acid (2.4%–9.0%), oleic acid (13.3%–41.8%), linoleic acid (38.1%–66.4%), and linolenic acid (4.2%–14.2%) (Appendix A). 17% of the screened mutants decreased in saturated fatty acid content when compared to the wild-type; while almost 13% of the screened mutants showed >30% oleic acid content (Figure 3). For polyunsaturated fatty acids, there were 33 mutants with a lower omega-6/omega-3 ratio when compared to the wild-type, of which the ratio of one mutant was less than 4. 

### 3.3. New Sources of Soybean Seed Composition Traits Identified by Forward Genetic Screening 

From 881 M3 lines with seed composition phenotypes, we selected at least five lines under each trait to advance to M4 generations for forward genetic screening (Figure 1B). A total of nine lines showed stable phenotypes in five seed composition traits as observed at the M3 and M4 generations (Figure 4). Two mutant lines, F142 and F145, presented a significant decrease of up to 33.0% (*p* < 0.001) in palmitic acid content compared to the wild-type. The stearic acid content in the mutant line F571 almost doubled (91.7% higher, *p* < 0.05) when compared to the wild-type. The average oleic acid content in the seed oil of F145, F159, F773, and F274 significantly increased by 91.6% (*p* < 0.01), 70.0% (*p* < 0.05), 62.5% (*p* < 0.05), and 62.3% (*p* < 0.05), respectively, when compared to the wild-type. Significant changes in polyunsaturated fatty acids (PUFAs) were observed in the F232 line for linoleic acid content (+7.5%, *p* < 0.01) and in the F368 line for linolenic acid content (−38.9%, *p* < 0.001). For the soybean meal traits, four mutant lines, F690, F788, F80, and F103, showed decreased stachyose content by 25.7%–38.2% when compared to the wild-type. A 58.8% reduction in sucrose content was also observed from the F690 line at the M4 generation (Table 1). Additionally, a mutant line, F264, with elevated protein content (10.9% higher than wild-type) was successfully identified and confirmed.

### 3.4. Correlation Analyses of Soybean Seed Composition Traits of TILLING Population

Correlation coefficients were calculated for each pair-wise combination of seed composition traits from M3 mutants (Table 2; Appendix A). The relationships between seed meal and fatty acid phenotypes were weak (*r* < 0.190) even though some of them were statistically significant. A highly significant correlation was observed between protein and oil content (*r* = −0.575, *p* < 0.001) while both protein and oil content lacked a correlation with carbohydrate content (Table 2; Figure 5A). Among seed carbohydrate profiles, sucrose content was positively correlated with raffinose (*r* = 0.265, *p* < 0.001) and stachyose content (*r* = 0.551, *p* < 0.001). A weak correlation (*r* = 0.228, *p* < 0.001) was observed between raffinose and stachyose content (Table 2). In contrast, the relationships among the five fatty acids were all statistically highly significant. As expected, the oleic acid content was strongly negatively correlated with linoleic acid (*r* = −0.962, *p* < 0.001) as well as linolenic acid content (*r* = −0.704, *p* < 0.001), whereas the correlation was positive between linoleic acid and linolenic acid content (*r* = 0.595, *p* < 0.001) (Table 2; Figure 5B). Additionally, palmitic acid content was negatively correlated with oleic acid content (*r* = −0.527, *p* < 0.001) but positively correlated with polyunsaturated fatty acids, linoleic acid (*r* = 0.420, *p* < 0.001), and linolenic acid content (*r* = 0.325, *p* < 0.001). Stearic acid showed relatively weak correlations with oleic acid (*r* = 0.119, *p* < 0.001), linoleic acid (*r* = −0.204, *p* < 0.001), and linolenic acid content (*r* = −0.134, *p* < 0.001), respectively (Table 2).

### 3.5. Correlation Analyses of Soybean Seed Composition Traits of Soybean Germplasm Lines

Correlation analyses were conducted with seed composition phenotypes from a total of the 103 lines representing wild, landrace, and elite soybean lines (Table 3, Appendix A). Sucrose content was positively correlated with raffinose content (*r* = 0.450, *p* < 0.001). Sucrose and oil content were positively correlated (*r* = 0.537, *p* < 0.001), and each was positively correlated with oleic acid content (*r* = 0.378, *p* < 0.001; *r* = 0.420, *p* < 0.001, respectively) but negatively correlated with palmitic acid content (*r* = −0.373, *p* < 0.001; *r* = −0.357, *p* < 0.001, respectively) and linolenic acid content (*r* = −0.272, *p* < 0.01; *r* = −0.579, *p* < 0.001, respectively). A weak correlation was observed between protein and oil content (*r* = −0.198, *p* < 0.05) as well as between sucrose and stachyose content (*r* = 0.105, *p* = 0.293) (Table 3). The correlation analysis among the five fatty acid contents revealed that oleic acid content was negatively correlated with linoleic acid content (*r* = −0.644, *p* < 0.001) and linolenic acid content (*r* = −0.788, *p* < 0.001). In addition, palmitic acid content was negatively correlated with oleic acid content (*r* = −0.388, *p* < 0.001), while the correlation between palmitic acid content and linolenic acid content was positive (*r* = 0.336, *p* < 0.001) (Table 3). 

## 4. Discussion

Soybean seed composition traits are the principal components of soybeans as a worldwide commodity with increasing demands. In 2018, genetically modified (GM) soybean accounted for 50% of the global GM crop production [40], however, the restricted policy for their commercialization and the consumer preference for GM-free products are prevailing throughout the world. In mutation breeding, chemical or physical mutagenesis provides an alternative strategy to produce large-scale soybean mutants for improving economically important traits [22]. It dramatically reduced the time needed for breeding soybean varieties with novel traits to meet the market’s demand. Employing forward and reverse genetics, a large number of induced soybean mutants became available for trait evaluations and gene functional studies [41]. In this study, a combined population of over 4000 soybean M2 and M3 lines were developed, from which M3 seeds from each mutant line were prepared for forward genetic screening. To cover the major soybean seed composition traits, seed protein, oil, carbohydrate, and fatty acid contents were measured for a subset of 881 M3 lines (Figure 2 and Figure 3). A broad range of contents in each of the ten traits was found from the current EMS mutagenized soybean population, even though they all shared the same genetic background. The seed content of the seven traits were similar among the set of 103 soybean accessions and the mutant population, including sucrose, stachyose, palmitic acid, stearic acid, oleic acid, linoleic acid, and linolenic acid content. The mutants presented a wide range of four fatty acids, including palmitic acid, stearic acid, oleic acid, and linoleic acid content. Narrower ranges of contents were only observed in raffinose, protein, and oil traits from the mutant lines (Appendix A). 

To test the trait heritability across generations, at least five lines from each trait were selected to advance to the M4 generation for forward genetic screening. This strategy successfully yielded a variety of mutants with high stearic acid or high oleic acid content from another EMS mutagenized soybean ‘Forrest’ population [32,42]. From this study, 14 mutant lines that showed stable phenotypes in seed meal and oil composition traits could become novel sources for breeding of desired traits (Figure 4, Table 1). Although we noted the existence of a negative correlation between protein and oil content in soybean seeds, we were able to identify a high protein (41.56%) mutant line (F264) while maintaining the wild-type oil (16.42%). Given the positive correlation between sucrose and stachyose content in the M3 mutants, a mutant (F690) was found to present both low sucrose and stachyose contents (Table 1). Several soybean lines have been reported to have >80% oleic acid and extremely low linoleic acid (<7%) when combining the mutant *FAD2-1A* and *FAD2-1B* alleles [43,44]. Considering the typical wild-type soybean genotype with low oleic acid/high linoleic acid phenotype, we were also able to identify four mutants with elevated oleic acid (30.9%–34.9%) as potential materials for crossing to reach the similarly high level of oleic acid reported in previous studies (Figure 4).

A comprehensive correlation analysis of M3 mutants in ten important seed composition traits revealed interesting findings when compared to previous studies. No positive correlation was found between protein and stachyose in the M3 population; thus, it is practical to develop high protein soybean lines with low stachyose using our mutant materials. Although yield data were not included for the correlation analysis, the previously reported negative correlation between protein and yield and the positive correlation between oil and yield indicated weak correlations among protein, oil, and yield traits due to the low correlation coefficient (*r* < 0.3) [17,37]. The weak correlation between protein and sucrose was also observed in the M3 population, while protein was repeatedly reported to negatively correlate with sucrose [19,45,46]. In the M3 population, the oil showed no or weak correlation with sucrose, stachyose, and raffinose, meaning the pathways converting carbohydrates to oil were severely affected by the chemical mutagen during the mutagenesis process. A new relationship was also observed in the M3 populations, where a positive correlation between sucrose and stachyose has never been reported before. For the correlation analysis of fatty acids, the negative correlation of oleic and linoleic acids as well as oleic and linolenic acids in the M3 population demonstrated the remarkable resilience of fatty acid biosynthetic pathways to EMS mutagenesis in soybeans [47].

Compared to the correlation analysis of the 103 soybean germplasm lines, the most noticeable difference was a lack of relationship between soybean meal and fatty acid traits in the M3 population. Sucrose was significantly correlated with oil, palmitic, oleic, and linolenic acids in the 103 soybean germplasm lines, but not in the M3 population (Table 2 and Table 3). Likewise, no or weak correlations were found in the relationships between oil and palmitic acid, oil and oleic acid, and oil and linolenic acid in the M3 population, in contrast with the 103 soybean germplasm lines. Such correlations were maintained through the evolution of cultivated soybean lines, even though spontaneous mutations occurred during domestication. However, induced mutations can cause rapid and widespread disruptions in major genes within the carbon cycle, therefore, these correlations turned into a weak or lack of correlation in the mutagenized populations. On the other hand, the correlation strength and direction within fatty acid content remained almost the same between the M3 population and the 103 soybean germplasm lines. The weak correlation between palmitic acid and stearic acid as well as stearic acid and oleic acid could be explained by the presence of various enzymes that may compete for the same substrate in the fatty acid biosynthetic pathway. For example, palmitic acid is not only converted into stearic acid that is catalyzed by the KASII enzyme but also transported out of the plastid as a free fatty acid by the FATB enzyme [11]. Therefore, stearic acid is not solely dependent on palmitic acid, and vice versa. Moreover, the strong correlation among oleic, linoleic, and linolenic acids could be largely due to the same subcellular localization of enzymes involved in their conversions within the endoplasmic reticulum, such as FAD2 and FAD3. The negative correlations obtained with linolenic acid are due to the fact that it is considered the last PUFA made from the fatty acid biosynthetic pathway in most of the plant lineages, including soybeans [48].

To the best of our knowledge, the majority of the correlation studies performed on the soybean seed composition traits were conducted based on the Pearson correlation coefficient estimations [49,50,51]. However, the normal distribution of two variables is an important prerequisite for using this method. If the data did not fit this distribution or outliers existed in datasets, the incorrect correlation coefficient could be calculated, causing misleading interpretations. For such skewed data including outliers, the nonparametric measure of association, named Spearman rank correlation, became a more appropriate alternative [52,53]. Before running the correlation analysis, we examined the distribution of all variables from the obtained data through a graphical method, normal quantile plot. The normally distributed data will resemble a straight diagonal line. Only a few variables in this study, such as protein, palmitic acid, and linolenic acid at the M3 generation, met the requirements for the Pearson correlation coefficient (Appendix A). Therefore, the Spearman rank correlation was used to produce more reliable and accurate results. Although the correlation results are similar in many cases between Spearman and Pearson, considerable discrepancies were found in soybean germplasm lines, such as *r* = 0.450 in Spearman vs. *r* = 0.188 in Pearson for correlation between sucrose and raffinose content and *r* = −0.106 in Spearman vs. *r* = −0.575 in Pearson for correlation between raffinose and stachyose content (Appendix A). This was mainly due to an outlier, PI (Plant Introduction) 603176A (HN096), included in the soybean germplasm dataset. This PI line showed the highest raffinose content (4.7%) and the lowest stachyose content (0.4%) among the 103 soybean accessions. These data demonstrated the negative effect of extreme values in the Pearson correlation. For interpreting correlation coefficients, another common misuse is to judge the strength of correlation only based on the general guideline of the absolute value of correlation coefficients. Nevertheless, the coefficient of determination (*r^2^*) is a more powerful parameter to explain the correlations [54]. For example, given the correlation coefficient was −0.962 between oleic acid and linoleic acid, 92.5% of the variation in oleic acid content could be explained by linoleic acid content and vice versa. On the contrary, 10.6% of the variation in palmitic acid content was explained by linolenic acid content.

Traditionally, the natural variations were heavily used in conventional soybean breeding programs to select desired traits. However, traditional plant breeding is encumbered with low frequencies of spontaneous mutations and an extended selection process to increase yield and quality traits [55]. In our current study, it was extremely difficult to select a soybean germplasm line with multiple desired traits, for example, a line with high protein, high sucrose, and low stachyose contents or a line with high sucrose, high oleic acid, low palmitic acid, and low linolenic acid contents. On the contrary, combinations of multiple traits can be easily identified from the individual EMS-mutant lines. In mutation breeding, genetic variations from induced mutations provide abundant sources of alterations in important soybean traits. Although many soybean mutants with improved seed composition traits are available, extra backcrossing is still needed to incorporate the desired traits into elite cultivars due to the background mutations caused by random mutagenesis. Therefore, development of new soybean varieties with improved seed composition traits can be achieved through a combination of natural and induced variations.

## Figures and Tables

**Figure 1 genes-10-00975-f001:**
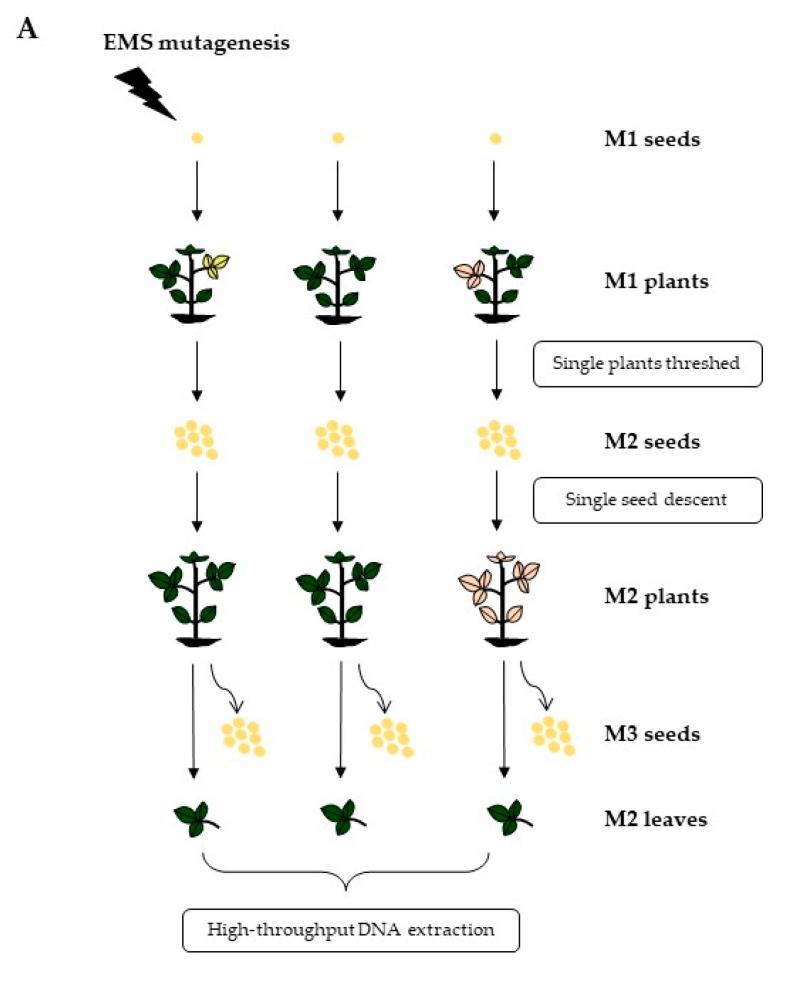
An overview of the TILLING (Targeting Induced Local Lesions IN Genomes) process in soybean cv ‘Forrest’. (**A**) Developing an ethyl methanesulfonate (EMS) mutagenized soybean population. A total of 2366 ‘Forrest’-derived M2 families have been developed; (**B**) Forward/reverse genetic screening for the M3 and M4 generations. A subset of M3 families were chosen to measure seed composition traits. Seeds from selected M3 lines were planted to advance to the M4 generation. Leaves from the M3 plants were used for genotyping. Forward rescreening was performed on the M4 seeds.

**Figure 2 genes-10-00975-f002:**
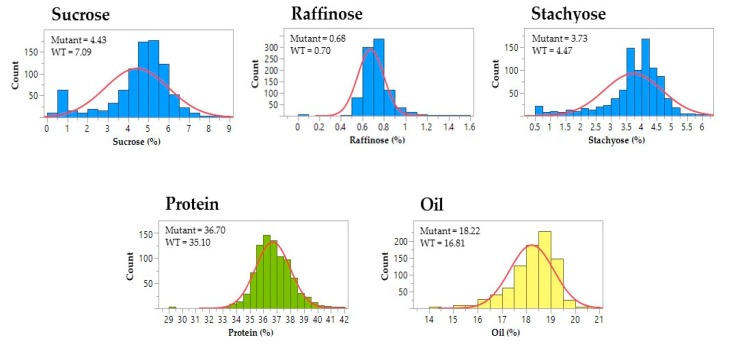
The distribution of seed meal phenotypes in the soybean M3 population. The histograms represent phenotypic variations in five seed meal traits among M3 lines. The means of sucrose, raffinose, stachyose, protein, and oil content are shown for the mutants and wild-type soybeans (WT), respectively. The blue curves indicate the trend of normal distribution.

**Figure 3 genes-10-00975-f003:**
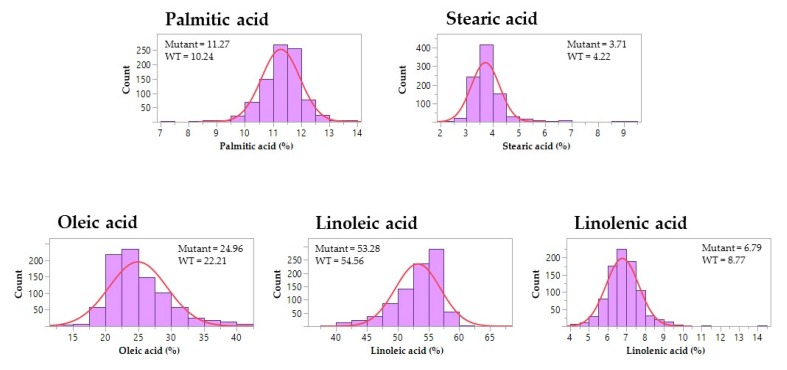
The distribution of seed fatty acid phenotypes in the soybean M3 population. The histograms represent phenotypic variations in five fatty acid traits among M3 lines. The means of palmitic acid, stearic acid, oleic acid, linoleic acid, and linolenic acid content are shown for the mutants and wild-type soybeans (WT), respectively. The blue curves indicate the trend of normal distribution.

**Figure 4 genes-10-00975-f004:**
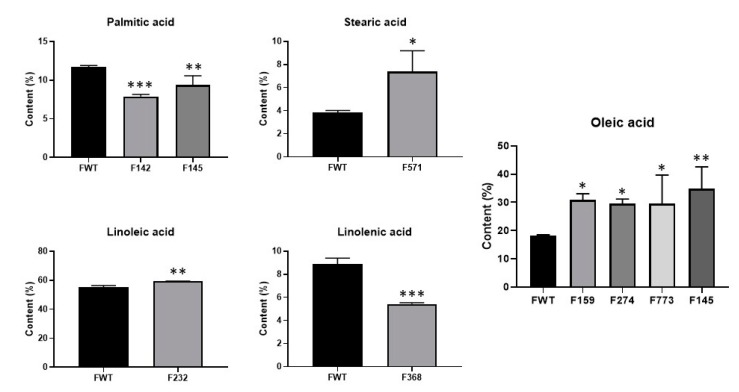
The fatty acid levels of the selected mutant lines in the M4 generation and ‘Forrest’ wild-type (FWT). Multiple pair-wise comparisons were performed on fatty acid contents between mutants and the ‘Forrest’ wild-type using Student’s t test. At least three replicates were tested for each line. Significance level: * *p* < 0.05, ** *p* < 0.01, *** *p* < 0.001.

**Figure 5 genes-10-00975-f005:**
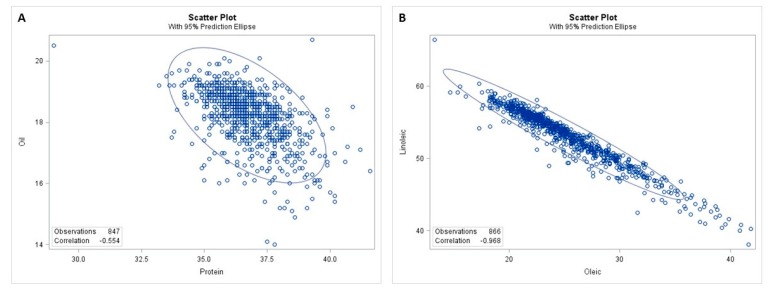
Examples of the scatterplots for correlation analysis in the M3 population. The two scatterplots showing the statistically significant correlations in the soybean M3 population with a 95% prediction ellipse. (**A**). protein-oil correlation; (**B**). oleic-linoleic correlation.

**Table 1 genes-10-00975-t001:** Seed meal phenotypes of the selected mutant lines (M3 and M4 generations) and ‘Forrest’ wild-type.

Plant ID	Sucrose (%)		Raffinose (%)		Stachyose (%)		Protein (%)		Oil (%)	
	M3	M4	M3	M4	M3	M4	M3	M4	M3	M4
F690	0.56	2.42	0.59	0.78	2.03	3.45	36.87	-	17.79	-
F788	0.67	4.52	0.67	0.81	2.05	3.15	37.13	-	17.20	-
F80	0.56	4.59	0.48	0.93	2.02	3.55	37.81	-	18.39	-
F103	0.71	5.58	0.49	1.00	1.62	3.79	36.82	-	18.30	-
F264	5.33	-	0.91	-	3.72	-	41.56	39.88	16.42	-
Forrest	7.09	5.88	0.70	0.81	4.47	5.10	35.10	35.95	16.81	-

Notes: Seeds from each line were measured in bulk.

**Table 2 genes-10-00975-t002:** Spearman correlation analyses of seed composition traits in the soybean M3 population.

	Sucrose	Raffinose	Stachyose	Protein	Oil	16:0	18:0	18:1	18:2	18:3
Sucrose										
Raffinose	0.265 ***									
Stachyose	0.551 ***	0.228 ***								
Protein	−0.117 ***	−0.011 ns	−0.067 ns							
Oil	0.011ns	−0.122 ***	0.020 ns	−0.575 ***						
16:0	0.028 ns	−0.028 ns	0.051 ns	−0.106 **	0.106 **					
18:0	0.033 ns	−0.079 *	0.045 ns	0.031 ns	−0.151 ***	−0.153 ***				
18:1	−0.021 ns	−0.014 ns	−0.052 ns	0.186 ***	−0.129 **	−0.527 ***	0.119 ***			
18:2	−0.004 ns	−0.008 ns	0.039 ns	−0.187 ***	0.161 ***	0.420 ***	−0.204 ***	−0.962 ***		
18:3	0.066 ns	0.069 *	0.038 ns	−0.111 **	0.003 ns	0.325 ***	−0.134 ***	−0.704 ***	0.595 ***	

* Significant at 0.05 probability level; ** Significant at 0.01 probability level; *** Significant at 0.001 probability level; ns, Nonsignificant (*p* > 0.05).

**Table 3 genes-10-00975-t003:** Spearman correlation analyses of seed composition traits in the 103 soybean germplasm lines.

	Sucrose	Raffinose	Stachyose	Protein	Oil	16:0	18:0	18:1	18:2	18:3
Sucrose										
Raffinose	0.450 ***									
Stachyose	0.105 ns	−0.106 ns								
Protein	−0.278 **	0.082 ns	−0.275 **							
Oil	0.537 ***	0.243 ns	−0.126 ns	−0.198 *						
16:0	−0.373 ***	−0.020 ns	0.067 ns	0.181 ns	−0.357 ***					
18:0	0.227 *	−0.178 ns	0.058 ns	0.024 ns	0.240 *	−0.147 ns				
18:1	0.378 ***	0.177 ns	−0.071 ns	−0.004 ns	0.420 ***	−0.388 ***	0.058 ns			
18:2	−0.014 ns	0.018 ns	−0.180 ns	−0.068 ns	0.151 ns	−0.076 ns	0.128 ns	−0.644 ***		
18:3	−0.272 **	−0.249 *	0.187 ns	−0.117 ns	−0.579 ***	0.336 ***	−0.172 *	−0.788 ***	0.213 *	

* Significant at 0.05 probability level; ** Significant at 0.01 probability level; *** Significant at 0.001 probability level; ns, Nonsignificant (*p* > 0.05).

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
