# Peer review of "Assessment of Phenotypic Variations and Correlation among Seed Composition Traits in Mutagenized Soybean Populations"

_genes, 2019, doi:10.3390/genes10120975_

Round 1

Reviewer 1 Report

This is a nice paper which may be helpful to breeders. Some comments follow.

l. 15 & 347 "conventional breeding"

sounds as though the authors consider EMS-based mutation breeding to represent a non-conventional breeding method. At least in Europe, breeding methods termed as "non-conventional" usually are associated with gene technology. So better do not mix up these definitions.

l. 23 either "analyses ... have been" or "analysis ... has been"

l. 78 "genetically modified crops"

I'm aware that currently, "genetically modified" is the term usually found in scientific papers as well as in legislative texts in the context of genetic engineering. However, since "genetically modified" principally applies to any modification of genetic material, please check whether you better want to use a linguistically more clear-cut term, e. g. "genetically engineered". I'm saying this because using the term "genetically modified" in European legislation has caused a lot of confusion in peoples' minds, with the result that the European Court of Justice decided chemical and X-ray mutants to belong to the group of GMO.

l. 88 "The objective ... was to develop..."

It would be helpful to the reader to give a short statement on why none of the existing mutant libraries (cf. preceding sentence) was used for the present study.

l. 90-94

This part seems to better fit to Discussion or/and Abstract rather than Intro.

l. 112-113

has already been described above

l. 117 sown(?)

Sections 2.2 & 2.3

Did you analyse seed-sample repetitions within mutants lines? if not, this should be addressed under Discussion.

l. 123 & elsewhere "Total oil and protein contents" (plural)

l. 150 capitalise "student's" (Student was the author's name or pseudonym)

l. 159 "Wild-type ... mutagenesis." Can be skipped (see M&M, 2.1).

Fig. 1: The legend of Fig. 1 can substantially be shortened by skipping the details already described under M&M

l. 165-167 "In the second ... DNA extraction." see M&M, 2.1

l. 176 or "mutagenized soybean populations"?

l. 178 (Figures 2 and 3) plural

l. 179, 180 contents

l. 187 better skip "Interestingly" and stick to mere description of results.

l. 196 "A total of nine lines" please indicate the generation (M4) of these lines

l. 197 & 270 "constant phenotypes"

not clear what you mean by "constant" phenotypes, since constant implies stability over time or generations (or treatments). maybe "uniform" phenotypes within M4 lines? ... or phenotypes (contents) that significantly deviate from the wild-type?

The notion "constant" seems to be not really consistent with the ranges within M4 lines shown in Fig. 4(?)

l. 227 better skip "Surprisingly" (you may discuss in chapter 4 why this is surprising)

Legends of Tables 2 & 3: please check if you really want to speak of "phenotypes" when simply contents of single components are addressed

l. 282-307

(only a suggestion) This section on statistics appears to blast the discussion which in the preceding section deals with seed composition contents and their correlations and picks up this topic again in l. 308. Maybe you want to place the statistics section elsewhere(?)

l. 284 maybe "If the data does not fit this distribution ... datasets, incorrect correlation coefficients might be calculated, causing misleading ..."

l. 360 link to Supplementaries: "Access denied" because of "unfit for youth"

(maybe I'd let my system admin sort this out)

l. 392, 393, 395, 399, etc. throughout the References

please check journal names and monograph titles (capitalise)

l. 401 do not abbreviate journal's name

That's it.

Reviewer 2 Report

Assessment of phenotypic variations and correlation among seed composition traits in mutagenized soybean populations is work about improvement of seed composition trait through mutation breeding. The article could be improved if authors considers following points

Absatrct should be presented in a quantitative way with rationale. In Introduction what is need of this study and which problems can be solved by this should be mentined clearly. Hypothesis needed Why only correlation was used what about use of PCA ? Discussions needs to be improved.
